# The adverse effects of trastuzumab-containing regimes as a therapy in breast cancer: A piggy-back systematic review and meta-analysis

Christopher Jackson[1], Leila Finikarides[2], Alexandra L. J. Freeman[2]*

1 MRC Biostatistics Unit, University of Cambridge, Cambridge, United Kingdom, 2 Winton Centre for Risk & Evidence Communication, Department of Pure Mathematics & Mathematical Statistics, University of Cambridge, Cambridge, United Kingdom

* Alex.freeman@maths.cam.ac.uk

**Data Availability Statement:** The data underlying the results presented in the study are available from https://doi.org/10.17605/OSF.IO/ER9FZ.

## Abstract

### Background

Trastuzumab is a valuable therapy option for women with ERBB2(HER2)+ breast cancer tumours, often used in combination with chemotherapy and alongside other therapies. It is known to have adverse effects, but these have proved difficult to separate from the effects of other concurrent therapies patients are usually taking. This study aims to assess the adverse effects specifically attributable to trastuzumab, and whether they vary by patient subgroup or concurrent therapies.

### Methods

As registered on PROSPERO (CRD42019146541), we used previous systematic reviews as well as the clinicaltrials.gov registry to identify randomised controlled trials in breast cancer which compared treatment regimes with and without trastuzumab. Neoadjuvant, adjuvant and metastatic settings were examined. Data was extracted from those which had, as of July 2022, reported adverse events. Risk of bias was assessed using ROB2. Primary outcomes were adverse events of any type or severity (excluding death). A standard random-effects meta-analysis was performed for each outcome independently. In order to ascertain whether adverse effects differed by individual factors such as age or tumour characteristics, or by use of trastuzumab concurrently with hormone therapy, we examined individual-level patient data for one large trial, HERA.

### Results

79 relevant trials were found, of which 20 contained comparable arms of trastuzumab-containing therapy and corresponding matched therapy without trastuzumab. This allowed a comparison of 8669 patients receiving trastuzumab versus 9556 receiving no trastuzumab, which gave a list of 25 statistically and clinically significant adverse effects related to trastuzumab alone: unspecified pain, asthenia, nasopharyngitis, skin disorders (mainly rash), dyspepsia, paraesthesia, infections (often respiratory), increased lacrimation, diarrhoea,

**Funding:** CJ was supported by the Medical Research Council, programme number MRC_MC_UU_00002/11.

**Competing interests:** The authors have declared that no competing interests exist.

myalgia, oedema (limb/peripheral), fever, nose bleeds, cardiac events, insomnia, cough, back pain, dyspnoea, chills, dizziness or vertigo, hypertension, congestive heart failure, increased levels of aspartate aminotransferase, gastrointestinal issues and dehydration. Analysis of individual patient-level data from 5102 patients suggested that nausea is slightly more likely for women taking trastuzumab who are ER+ /also taking hormone therapy than for those who are ER-/not taking hormone therapy; no other potential treatment-subgroup interactions were detected. We found no evidence for significantly increased rates of neutropenia, anaemia or lymphopenia in patients on trastuzumab-containing regimes compared to those on comparable regimes without trastuzumab.

## Conclusions

This meta-analysis should allow clinicians and patients to better identify and quantify the potential adverse effects of adding trastuzumab to their treatment regime for breast cancer, and hence inform their decision-making. However, limitations include serious risk of bias due to heterogeneity in reporting of the outcomes and the open-label nature of the trials.

## Introduction

Patients and clinicians face difficult choices when planning treatment regimes, particularly when the options come with significant adverse effects. Tools such as Predict:Breast Cancer (breast.predict.nhs.uk) present assessments of the potential benefits of treatment options, but the potential adverse effects of those options have not been collated and quantified in the same way, making comparisons of the risk and benefit of each option difficult. We undertook this study as the result of consultation with breast cancer patients on what information they wanted when making treatment decisions.

Trastuzumab ('Herceptin') is a monoclonal antibody, created to inhibit the growth of cells that have an overexpression of the ERBB2 (previously HER2) gene. The products of this gene can cause increased sensitivity of cells to epidermal growth factors, causing malignancy. Tumours that overexpress the ERBB2 gene–around 15–30% of human breast cancers—are typically aggressive and difficult to treat. Trastuzumab was shown in 1996 to be able to help inhibit growth of these cells in breast cancer patients [1]. Since then, large randomised controlled trials and meta-analyses have shown trastuzumab to delay the recurrence of breast cancer and increase survival for women who have ERBB2 positive tumours [2]. Since the development of trastuzumab, trastuzumab-resistance in tumours has been noted and trastuzumab has been combined with the cytotoxic drug DM1 to form trastuzumab emtansine (T-DM1), and it is commonly used in combination with other ERBB2-targeted therapies such as lapatinib and pertuzumab, or mTOR inhibitors such as everolimus, and work has been ongoing on understanding its mechanism of action [3]. In addition, trastuzumab (and subsequent biosimilar drugs) have been licenced for use not just for patients with ERRB2 positive breast cancer, but also gastric cancer [4]; and its derivatives show promise in other cancers such as ERRB2 positive non-small-cell lung cancer [5] and are being tested in others [6].

Humanized monoclonal antibodies have been associated with side effects involving an immune system response (such as anaphylaxis on first administration), haematological toxicity (especially when used alongside chemotherapy) causing neutropenia or lymphopenia and a resulting increased risk of infection (or symptoms from a prior infection such as CMV), and

cardiac or pulmonary complications [7]. Epidermal growth factor inhibitors, including trastuzumab, have also been associated with skin and nail changes such as rashes [8].

The cardiac effects of trastuzumab, and particularly the concurrent or sequential use of trastuzumab with or after chemotherapy has received particular attention [9–11]. One early retrospective analysis of the cardiac effects identified that 3–7% of patients receiving trastuzumab alone developed cardiac dysfunction, but that this rose to 13% of those receiving paclitaxel and trastuzumab, and 27% of those receiving and anthracycline plus cyclophosphamide as well as trastuzumab [11]. A later meta-analysis of the use of trastuzumab in early breast cancer included 8 trials and looked at the risk of damage to the heart and haematological toxicity due to trastuzumab and found an increased risk of congestive heart failure and decreased left ventricular ejection fraction [2]. A similar meta-analysis of the use of trastuzumab in metastatic breast cancer included 7 trials and concluded that for every 1000 women with metastatic breast cancer not taking trastuzumab, 10 would have heart toxicities (taking other anti-cancer therapies), whilst for every 1000 of those taking trastuzumab, an extra 25 would suffer severe heart toxicities [12]. Clinical trials now routinely monitor cardiac function and adjust patients' individual treatments if problems are identified.

The same authors in their reviews found a slight increase in the risk of neutropenia in women with metastatic breast cancer taking trastuzumab, but no evidence of increased risk of neutropenic fever or haematological toxicities. They did not find enough data to assess the effect on the other effects they investigated (brain metastases, treatment-related death, quality of life) [12]. A narrative review highlighted febrile neutropenia, diarrhoea and fatigue as potential side effects of the drug when looking at grade 3 or 4 toxicities [10].

The Summary of Product Characteristics for trastuzumab lists: infections and nasopharyngitis; anaemia, neutropenia, febrile neutropenia, leukopenia and thrombocytopenia; anorexia; insomnia; tremor, dizziness, headache, paraesthesia, dysgeusia, peripheral neuropathy; increased lacrimation and conjunctivitis; increased and decreased blood pressure, irregular heartbeat, decreased ejection fraction; hot flushes; dyspnoea, cough, epistaxis, rhinorrhoea; diarrhoea, vomiting, nausea, lip swelling, abdominal pain, dyspepsia, constipation, stomatitis; erythema, rash, facial swelling, nail disorder, alopecia, hand-foot syndrome; arthralgia, muscle tightness and myalgia; asthenia, chest pain, chills, fatigue, influenza-type symptoms, pain, infusion-related reaction, pyrexia, mucosal inflammation and peripheral oedema all as very common (>10 in 100 patients) in patients receiving either trastuzumab monotherapy, or in combination with chemotherapy. It is not clear which of these are adverse effects of trastuzumab alone. It is noted that about 40% of patients experience a mild to moderate acute reaction to the infusion, consisting of chills, fever, dyspnoea, hypotension, wheezing, bronchospasm, tachycardia, reduced oxygen saturation, rash, nausea, vomiting and headache [9].

A systematic review of the incidence of adverse effects of anti-ERBB2 therapies was published in 2015, which reported the adverse events described as treatment-related from seven trials of trastuzumab monotherapy where patients were receiving no other concurrent therapies [13]. They did not compare the incidences in the trastuzumab-treated arm with those in a control arm. This review identified gastrointestinal effects, skin rash, fatigue & weakness, joint pain and musculoskeletal pain, lip numbness, headache, chills, infection, depression, hot flushes, cardiac effects, hepatic failure, neutropenia, and hypertension as possible adverse effects.

We carried out this study to update and deepen the systematic approach to the identification and quantification of adverse effects for trastuzumab, sourcing data from as many patients as possible (and including patient-level data from one large trial) within randomised controlled trials in which a treatment regime containing trastuzumab was compared against a similar regime excluding trastuzumab. We also aimed to analyse data on a large number of possible adverse events, in order to determine:

For women with breast cancer receiving trastuzumab monotherapy as an adjuvant or neoadjuvant therapy:

- What adverse events are they more likely to experience, and with what relative frequency (by comparison with those not receiving trastuzumab)?

- What is the likely severity of those adverse events?

- Does the relative risk of events vary by dose, length of use, or other adjuvant therapy being received at the same time (e.g. concurrent chemotherapy or in combination with other ERBB2-targetting therapies)?

- Does the relative risk of events vary by patient characteristics such as age, menopausal status, or oestrogen or progesterone receptor status?

Given the existence of several recent, thorough and well-recorded systematic reviews of trials with these criteria in order to assess the benefits of trastuzumab in this patient population, in a similar method to one used by others previously [14, 15], we used these reviews as the major source of identified relevant trials, and hence call our work a 'piggyback' systematic review.

## Materials and methods

### Data sourcing

The original protocol for the literature review was registered on PROSPERO, registration number CRD42019146541, and can be accessed there, although we modified this by adding a search of the clinicaltrials.gov registry. We started by conducting an initial search for relevant systematic reviews. We searched PubMed, and Web of Knowledge using the following search terms: systematic AND breast AND trastuzumab, with the 'systematic' filter on in PubMed. The search was done for all time periods up until July 2022, with no restriction on publication language. Titles only were initially checked against the inclusion criteria by one author (AF): systematic reviews, where the population was patients with breast cancer, at least some of the trials were randomised, and had at least one arm containing trastuzumab. The full text of relevant-seeming titles were checked against the same selection criteria. References for all the randomised trials with at least one arm containing trastuzumab that were included in each of these systematic reviews were extracted. In order to supplement the trials cited by other meta-analyses with any more recently reporting trials, the registry clinicaltrials.gov was searched for all trials mentioning trastuzumab in the intervention, and with results available. This list was then filtered down to those classified as having a randomised, parallel arm study design. Trials were excluded if they were trials of trastuzumab biosimilars or compared methods of trastuzumab delivery. After de-duplication of these lists, the adverse events data reported from all trials was sought from academic papers, clinical trial reports, or clinical trials registries up until July 2022. From these trials, for a simple meta-analysis, we then sought those trials where adverse events were reported from a pair of arms in which the treatment regimes were the same apart from the presence/absence of trastuzumab. We allowed the inclusion of trials in which the arm not given trastuzumab were ERBB2 negative and the arm given trastuzumab were ERBB2 positive as long as the baseline characteristics were otherwise comparable.

### Additional data sources

In order to investigate the effect of patient characteristics on adverse events, individual patient-level data was sought from the large trial HERA (BIG 1–01; NCT00045032; European

Clinical Trials Database number 2005-002385-11) [16]. This trial randomised 5102 women in 39 countries after completion of primary therapy to one of three groups: no trastuzumab, 1 year of trastuzumab or 2 years of trastuzumab.

## Data extraction

A total of 22 trials was found where the study population were women with breast cancer, the protocol was randomised to at least 2 arms; where patients in one arm were given a regime or combination of regimes that involved trastuzumab and those in another arm the same regime or combination of regimes without trastuzumab; and where adverse events were numerically reported for the individual arms [17–50]. These trials together contained 26 paired arms (where trials had two pairs of arms, each with a comparable trastuzumab and non-trastuzumab regime, the pairs were treated separately in the analysis): 8669 patients receiving trastuzumab, versus 9565 receiving no trastuzumab. We further excluded trials which compared a dual therapy regime combining trastuzumab with another anti-ERBB2 therapy with the other anti-ERBB2 therapy alone, since the additional effects of trastuzumab may be different for women already taking other ERBB2 therapies. Efficacy and safety of such combination therapies has been previously studied by, (e.g., Kümler, Tuxen and Nielsen, 2014 [51]; Wilcken *et al.*, 2014 [10]; Ma *et al.*, 2019 [52]; Wu *et al.*, 2019 [53]).

All numerical information on adverse effects were extracted. Where percentages of patients affected were reported in papers, the actual numbers were back-calculated as closely as possible (rounding to the nearest whole number). Wherever possible, figures were used for the total number of patients actually treated per protocol (e.g. the 'safety population') rather than those initially randomised to it (intention to treat). Data from a trials registry were used wherever possible as these were the most detailed and complete.

Adverse events were often reported according to different scales and to different levels of detail. Some adverse events that were reported singly in some trials were clustered for recording (e.g. 'infections' to collate multiple kinds of miscellaneous infections, and 'skin effects' for multiple effects on the skin). This means that for these events, if a single individual suffered 3 different kinds of infection, say, it would be recorded as 3 people experiencing 'infection'. Data on the seriousness of the adverse effects was also collected where available.

Data was additionally collected on: funding source of trial, blinding, length of follow-up, therapy setting (adjuvant or neoadjuvant, metastatic or not), dose and regime of delivery, additional treatments the patients received, withdrawals from the trial due to adverse events. These are available in the dataset (made publicly available).

The database searching, trial selection and initial data extraction was done by one reviewer (AF) and the data extraction then independently checked by another (LF).

## Risk of bias

There were multiple sources of potential bias across the study, including major sources not included in standard risk of bias tools. Adverse events are usually only reported by the clinicians, rather than by the patients, and sometimes this only includes events thought by the clinician to be (likely) related to the treatment. The events are then sometimes recorded systematically, and sometimes not, and some reporting is only of selected adverse events. One author (AF) carried out a risk of bias assessment using the ROB2 checklist and spreadsheet [54]. We additionally report some potential sources of bias (such as source of funding, blinding, and number of patients lost to follow-up), in the dataset to allow readers to draw their own conclusions about the data quality.

## Analysis

Each of the 81 distinct types of recorded adverse effects were analysed as a separate outcome.

### Standard meta-analysis of the data in 26 paired arms: Trastuzumab vs control

To investigate what adverse effects breast cancer patients receiving trastuzumab are more likely to experience, and with what frequency (compared to those not taking trastuzumab), a standard random-effects meta-analysis was performed for each adverse effect independently. This was carried out on the 26 pairs of arms that directly compared patients taking trastuzumab with those on a similar regime but not taking trastuzumab. The odds ratio and risk difference for the binary symptom outcome, compared between trastuzumab and no-trastuzumab, was estimated for each pair of arms. Pooled odds ratios and risk differences were computed using random effects estimates based on the Mantel-Haenszel method, through the "meta" R package [55].

For each trial, the dose of trastuzumab given was recorded: either high (loading dose of 8mg/kg and 3-weekly 6mg/kg thereafter) or low (4mg/kg and 1-weekly 2mg/kg). In order to investigate dose-dependent differences in adverse effects, a descriptive comparison was made of the odds ratios and risk differences (for the adverse effect of trastuzumab) between high and low doses. The effect was also compared between trial arms that included only patients with metastatic cancer (8 arms) and the remaining 16 arms, 15 of which included only non-metastatic patients. There was not enough data to carry out formal meta-regression or any other subgroup analyses. Therefore we obtained individual patient-level data to answer our questions about adverse effect modifiers.

### Analysis of individual patient-level data in the HERA trial

In order to investigate whether patient characteristics, length of regime, or the presence of concurrent therapy affects the likely adverse effects profile of trastuzumab, individual patient-level data from the large HERA trial was used. The trial included patients with ERBB2-positive breast cancer who had completed initial tumour excision and at least four cycles of neoadjuvant or adjuvant chemotherapy. Patients continued other adjuvant therapies (such as hormone therapy or the taxane portion of their chemotherapy), but were randomised to three arms regarding trastuzumab therapy: none, 1 year of trastuzumab, or 2 years of trastuzumab. Data was available for each patient on menopausal status, oestrogen receptor (ER) status, progesterone receptor (PR) status, whether they took hormone therapy during the study, or whether they were still taking taxane chemotherapy.

The dataset included the 5099 patients in the safety analysis of the HERA trial. 31717 adverse events were observed, over 4162 patients. There were 1667 differently-named events in 26 different clinical classes. The dataset also contained serious adverse event indicators, durations, initial/extreme intensity of event, treatment given for adverse symptom, trial medication adjustment, outcome and CTC grade.

The data also recorded whether the adverse event was judged to be related to the trial medication, however, given the potential for bias in this judgement, this information was not used.

Firstly, the proportions of patients who experienced an event of each type were compared between the treatment groups, calculating the relative risk and risk difference between each trastuzumab group and control with 95% confidence intervals.

Secondly, we investigated whether the relative risk of the event between trastuzumab and control is modified by age, menopausal status, oestrogen receptor status, progesterone

receptor status, race, additional hormone therapy given, or type of chemotherapy given, by using logistic regression with a treatment-subgroup interaction.

## Results

The initial search for relevant systematic reviews produced a total of 124 titles on Web of Knowledge and 134 on PubMed. Selection based on titles left resulted in 68 systematic reviews being obtained at full text level. These reviews referenced 92 randomised controlled trials involving trastuzumab either as a monotherapy, as trastuzumab emtansine (T-DM1) or in combination with various other (neo)adjuvant regimes. A search and filtering of trials with results available on clinicaltrials.gov resulted in an initial list of 90 being identified. After de-duplication and exclusions of trials of biosimilars and modes of delivery, there were 110 potentially useful trials. Further exclusions were then made of trials which were dose escalation trials, involved combining or comparing trastuzumab with other experimental drugs, where adverse data was not reported in such a way as to be useful, or where the arms of the trial were not comparable in some way (other than ERBB2 status, as mentioned above). This left 79 trials. Fig 1 indicates the flow chart of study inclusion.

These 79 trials mostly fell into four categories: trials where trastuzumab was kept constant whilst the accompanying regime was altered; trials where the accompanying regime was the same but the trastuzumab was either not given to one arm (sometimes another ERBB2-targeting drug was used as a replacement), or was combined with another ERBB2-targeting drug in a dual therapy; trials where different trastuzumab regimes (such as length or dose) were tested; trials testing trastuzumab emtansine (T-DM1)–usually against trastuzumab. Some trials had multiple arms testing more than one of these potential combinations. In order to simplify the analysis to the most powerful and useful data, only the 22 trials which included arms where two comparable arms were given the same regime with or without trastuzumab were included. There were 26 such paired arms in these 22 trials. See S1 File for a summary of the characteristics of the trials identified and see Fig 2 for the risk of bias assessment of these 22 trials. The most frequent causes of potential bias were large numbers of patients discontinuing therapies, clinician-collected adverse event data in open-label trials, and lack of clarity in which adverse events were recorded and which not recorded or reported.

### Meta-analysis: Trastuzumab vs control

The left panel of Fig 3 shows, from the commonest at the top to the rarest at the bottom, the raw frequencies of each of the 79 adverse events (across all arms) which were reported in at least one study, over the pooled data from the 26 paired arms comparing patients whose treatment contained trastuzumab with those whose treatment did not. The right panel of Fig 3 shows the corresponding pooled estimates of the odds ratio from the random-effects meta-analysis, comparing the risks of suffering each adverse event in those whose treatment contained trastuzumab with the risk in those whose treatment did not contain trastuzumab. Events where there is a practically large and statistically significant difference in risk (difference $>0.02$) or in odds ratio ($>1.5$) are highlighted. There are 25 of these significant adverse events–listed here in order of commonness for patients: unspecified pain, asthenia, nasopharyngitis, skin disorders (mainly rash), dyspepsia, paraesthesia, infections (these were often respiratory), increased lacrimation, diarrhoea, myalgia, oedema (limb/peripheral), fever, nose bleeds, cardiac events, insomnia, cough, back pain, dyspnoea, chills, dizziness or vertigo, hypertension, congestive heart failure, increase levels of aspartate aminotransferase, gastrointestinal issues and dehydration.

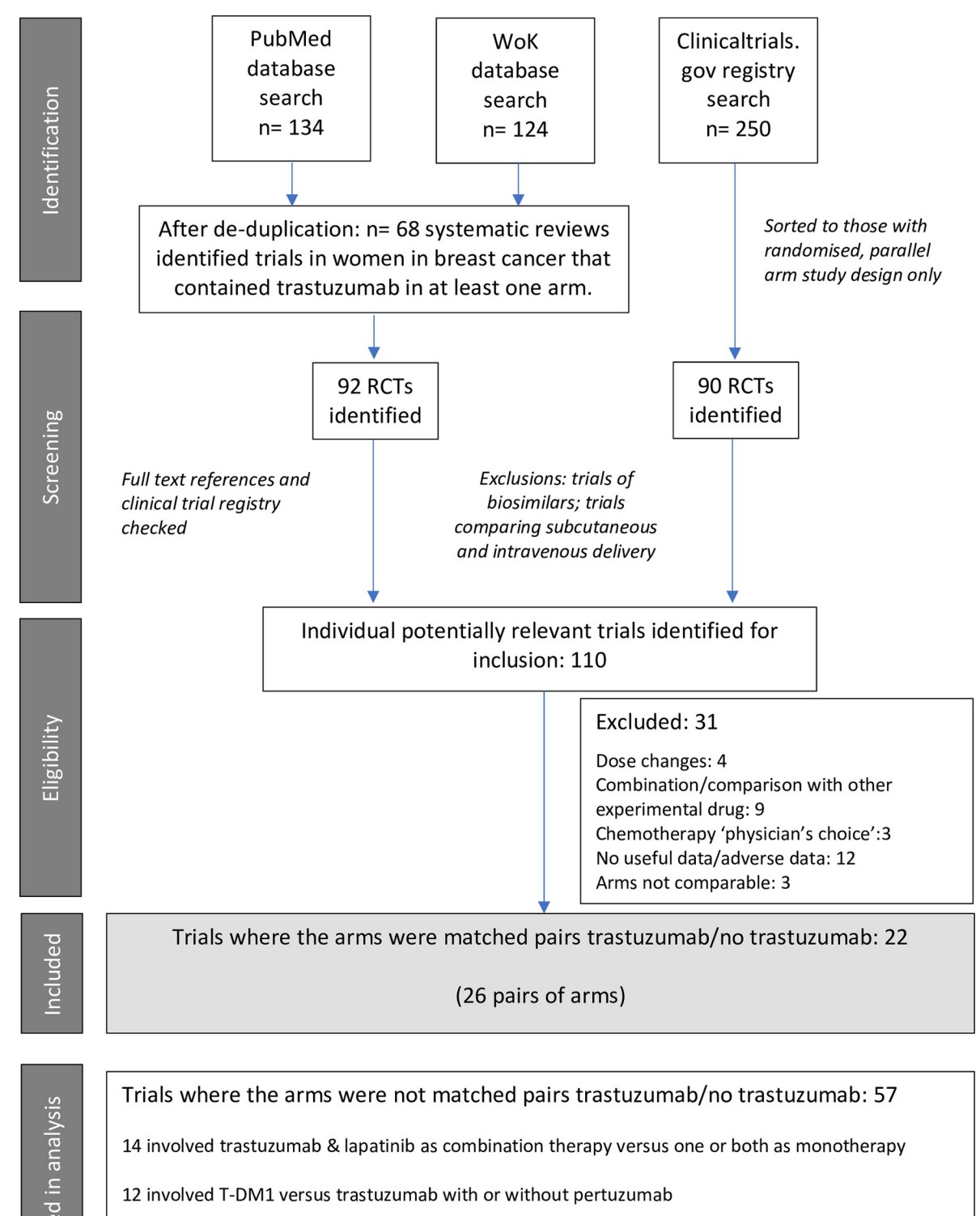

**Fig 1. PRISMA flow diagram for meta-analysis data collection.**

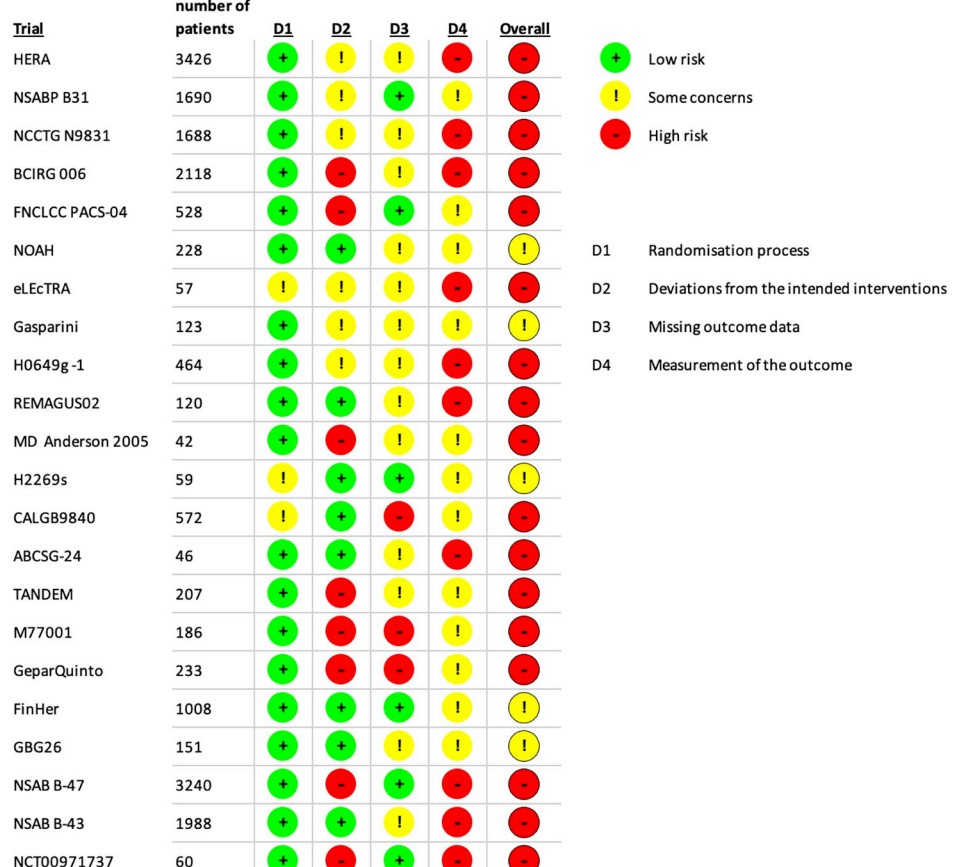

**Fig 2. Risk of bias assessment for the 20 trials included in the meta-analysis.**

The pooled odds ratios and risk differences for the events with significant effects of trastuzumab are presented in Table 1, which also shows the proportion of observed events that were described as Serious or Grade 3+ (depending on the classification system used for reporting). To put the additional risk of events given by trastuzumab into perspective, Table 1 also presents a crude estimate of the baseline risk of each event, obtained from meta-analysis of the control data from the studies that compared trastuzumab with a non-trastuzumab control (these control groups were also receiving other treatments for their cancer). However, given the incompleteness of adverse event reporting, and the variations between the study populations and the treatments they were receiving, the baseline risks in real populations will not necessarily be the same as this crude value.

The study-specific odds ratios for each significant event are illustrated in Fig 4. As in previous meta-analyses of adverse effects [56] there was considerable heterogeneity between studies in the relative risks of events between treatment groups. Some of this is likely due to different thresholds of recording or reporting (some only reporting Serious or Grade 3+ symptoms), but could also be due to the different treatment regimes experienced by the patients in different trials (e.g. chemotherapy and hormone regimes or trastuzumab dosing).

In order to investigate the possibility of different dosing affecting symptoms, we distinguished the study-specific odds ratios in Fig 4 by the different dosing regimes used. 12 studies

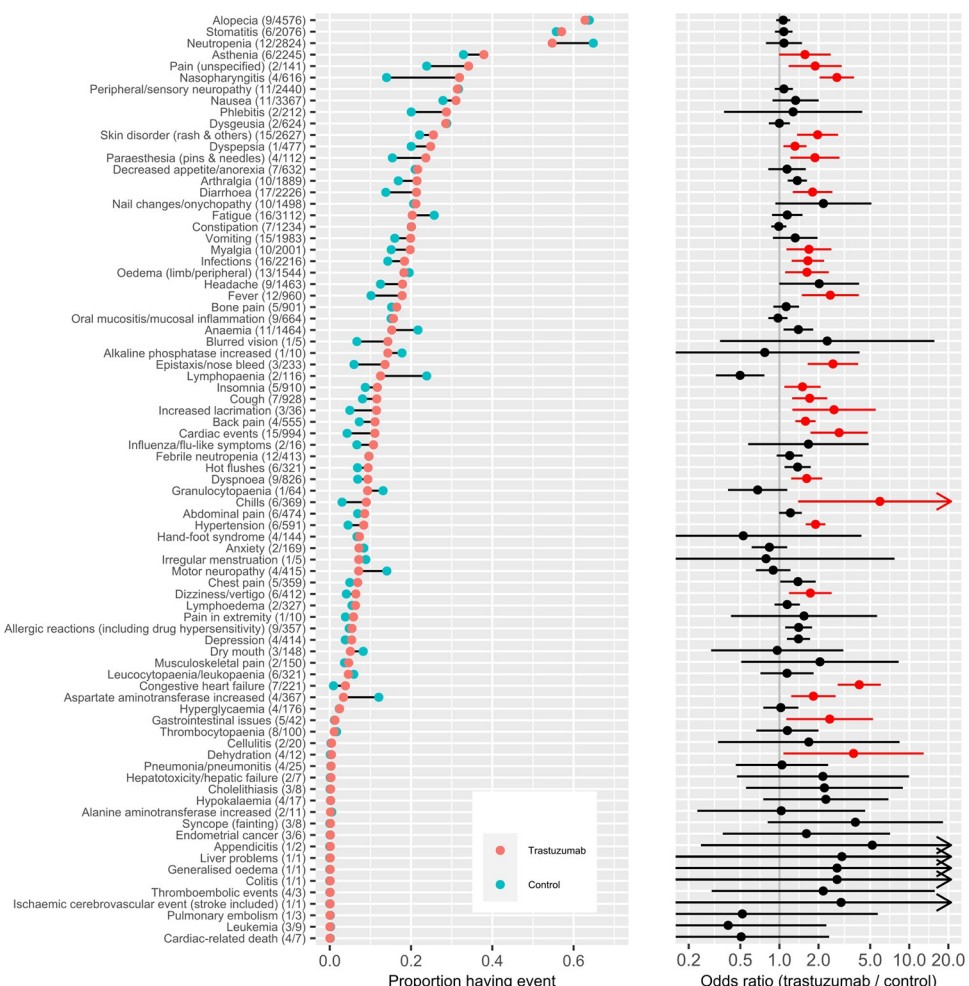

**Fig 3.** (left) Proportions of people in trastuzumab and control groups experiencing each adverse event, from pooled data from 23 trials. (right) Pooled odds ratio from random effects meta-analyses of each corresponding adverse effect of trastuzumab, with 95% confidence intervals (upper limit truncated at 20). Brackets after event name show (number of studies / overall number of observed events) contributing to each estimate.

used weekly low doses of trastuzumab (maintenance dose of 2mg/kg), and 11 used 3-weekly higher dosing (maintenance dose of 6mg/kg), both of which are used in clinical practice [9]. One (NSAB B-47) used two different doses for different patient sub-groups. No systematic difference in the odds ratios between the low dose and high dose trials was evident, though there were an insufficient number of studies to enable a formal meta-regression.

One small trial (H0650g –excluded from the main meta-analysis because it did not contain a non-trastuzumab control group) specifically tested a low weekly dose versus a mid-level weekly dose (maintenance dose of 4mg/kg) [57]. In this study, illustrated in Fig 5, most adverse events were more frequent in the medium dose group compared to the low dose group, though the uncertainty around the odds ratio was high for all events.

Similarly, no systematic difference could be detected in the event rates between studies of only patients with metastatic cancer (9 out of 26 studies), and studies of non-metastatic patients (not illustrated), though again, there were insufficiently many studies to allow a formal meta-regression analysis.

**Table 1. Events for which there was a statistically and practically significant (estimated odds ratio 1.5 or risk difference 2%) adverse effect of trastuzumab, from the meta-analysis.** Estimates of odds ratio and risk difference given with 95% confidence intervals.

| Outcome | Odds ratio | Crude estimate of the baseline risk (number out of 100 expected to experience the event if not taking trastuzumab) | Number of *additional* women out of 100 who would experience the event if taking trastuzumab | Percentage of graded events that were reported as Serious or Grade 3+ (absolute numbers) | Number of studies reporting this event (number of women reporting event / total number of women) |
|---|---|---|---|---|---|
| Aspartate aminotransferase increased | 1.83 (1.23–2.71) | 6 | 6 (-2-15) | 1.9% (7/367) | 4 (367/4307) |
| Asthenia | 1.57 (1.00–2.48) | 20 | 3 (1–4) | 2.6% (58/2245) | 6 (2245/6338) |
| Back pain | 1.59 (1.33–1.9) | 9 | 4 (2–5) | 2.9% (16/555) | 4 (555/6029) |
| Cardiac events | 2.88 (1.73–4.79) | 4 | 7 (3–12) | 27.0% (291/1077) | 15 (994/12940) |
| Chills | 5.94 (1.40–25.24) | 3 | 11 (3–19) | 2.7% (10/369) | 6 (369/6148) |
| Congestive heart failure | 4.12 (2.81–6.04) | 1 | 2 (1–4) | 48.7% (95/195) | 6 (195/7606) |
| Cough | 1.71 (1.25–2.33) | 8 | 3 (2–4) | 0.2% (2/922) | 7 (928/9449) |
| Dehydration | 3.72 (1.07–12.91) | <1 | <1 | 75% (9/12) | 4 (12/4372) |
| Diarrhoea | 1.80 (1.27–2.56) | 10 | 5 (2–7) | 7.9% (176/2220) | 16 (2137/11058) |
| Dizziness/vertigo | 1.73 (1.19–2.52) | 3 | 2 (0–4) | 0.7% (3/412) | 6 (412/7858) |
| Dyspepsia | 1.32 (1.07–1.62) | 20 | 5 (1–8) | 0% (0/477) | 1 (477/2118) |
| Dyspnoea | 1.62 (1.23–2.13) | 7 | 2 (0–4) | 7.4% (61/826) | 9 (826/10174) |
| Epistaxis | 2.58 (1.65–4.04) | 6 | 8 (1–16) | 0% (0/233) | 3 (233/2363) |
| Fever | 2.47 (1.49–4.09) | 7 | 7 (3–11) | 23.6% (227/960) | 12 (960/6898) |
| Gastrointestinal issues | 2.44 (1.13–5.26) | 3 | 4 (-2-10) | 51.3% (20/39) | 5 (42/3462) |
| Hypertension | 1.9 (1.59–2.26) | 4 | 3 (1–5) | 1.7% (10/591) | 6 (591/9224) |
| Increased lacrimation | 2.63 (1.26–5.51) | 5 | 6 (-3-14) | 2.8% (1/36) | 3 (36/452) |

(*Continued*)

**Table 1.** (*Continued*)

| Outcome | Odds ratio | Crude estimate of the baseline risk (number out of 100 expected to experience the event if not taking trastuzumab) | Number of *additional* women out of 100 who would experience the event if taking trastuzumab | Percentage of graded events that were reported as Serious or Grade 3+ (absolute numbers) | Number of studies reporting this event (number of women reporting event / total number of women) |
|---|---|---|---|---|---|
| Infections | 1.66 (1.24–2.21) | 6 | 4 (1–6) | 12.0% (267/2216) | 16 (2216/13678) |
| Insomnia | 1.5 (1.09–2.08) | 8 | 3 (1–4) | 0% (0/910) | 5 (910/8903) |
| Myalgia | 1.69 (1.13–2.52) | 9 | 3 (1–6) | 0.6% (12/1996) | 9 (1833/9800) |
| Nasopharyngitis | 2.77 (2.04–3.76) | 15 | 16 (12–21) | 0% (0/616) | 4 (616/2662) |
| Oedema (limb/peripheral) | 1.63 (1.11–2.40) | 10 | 5 (1–10) | 2.3% (36/1538) | 13 (1544/8140) |
| Pain (unspecified) | 1.89 (1.18–3.02) | 6 | 7 (-5-20) | 13.5% (19/141) | 2 (141/485) |
| Paraesthesia (pins & needles) | 1.88 (1.21–2.9) | 16 | 8 (2–14) | 1.9% (2/108) | 4 (112/580) |
| Skin disorder (rash & others) | 1.97 (1.37–2.83) | 10 | 5 (1–8) | 1.7% (44/2627) | 15 (2627/11104) |

## Analysis of individual patient-level data from HERA

The data from the HERA trials made available to us allowed us to look at differences in adverse event frequencies between patients given 1 year or 2 years' worth of trastuzumab therapy in comparison with control. Fig 6 shows the events for which the relative risk or risk difference between either 1 year or 2 years of trastuzumab and control was both statistically significant ($p<0.05$) and clinically significant (defined by either a relative risk of 1.5 or more, or a risk difference of 2% or more).

These 13 significant adverse events reflect, in general, those found in the simple meta-analysis as well: effects on the skin and nails, the heart, infections (especially respiratory infections), gastrointestinal effects, muscle spasms, mucosal inflammation and nasal dryness. There are some omissions, such as pyrexia and chills, which did not show up as significant in the analysis of HERA. It is possible that these relate to interactions between trastuzumab and concurrent chemotherapy, since in the HERA trial, trastuzumab was mainly given after the completion of chemotherapy. Alternatively, this may be due to the more limited power of a single trial to detect these effects.

The advantage of patient-level data is that it allows subgroup analyses to investigate how the effect of trastuzumab is modified by other drugs taken concurrently, or by patient characteristics.

With the data from HERA, we were able to look at age, menopausal status, oestrogen receptor status (ER), progesterone receptor status (PR) and race as potential adverse effect modifiers. Table 2 shows the only effect—nausea—for which there was a statistically significant treatment-subgroup interaction at the 5% level (considering treatment group as a 3-level

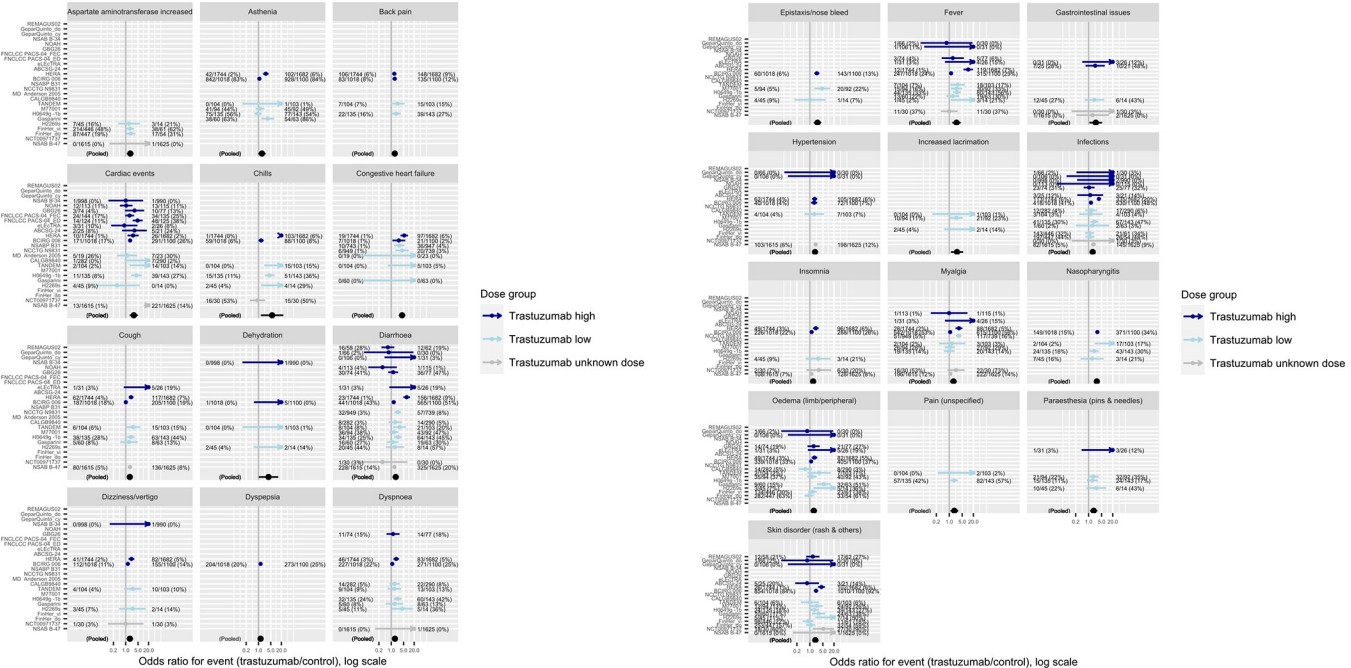

**Fig 4.** a. Study-specific odds ratios for 12 out of the 25 events for which a statistically and practically significant adverse of effect of trastuzumab was determined (the remaining 12 shown in Fig 4b. b. Study-specific odds ratios for 13 out of the 25 events for which a statistically and practically significant adverse of effect of trastuzumab was determined (the remaining 12 shown in Fig 4a). Trastuzumab dose highlighted in colour, and estimates shown with 95% confidence intervals.

categorical variable). It shows the ratio between odds ratios (for the effect of 2 years trastuzumab on event rates) between subgroups, the odds ratios with and without the risk factor defined by the subgroup, for each of 1 year and 2 years trastuzumab, and a p-value for the likelihood ratio test of interaction. This suggests that nausea is slightly more likely for women taking trastuzumab who are ER+ /also taking hormone therapy than for those who are ER-/not taking hormone therapy. However, no other interactions were detected as significant.

## Conclusions

To provide the most reliable estimates of the side effects of trastuzumab as a monotherapy, we have combined data from 18,234 patients in 26 randomised trial arms, covering 79 potential adverse effects. Through meta-analysis, we have identified 25 adverse effects whose frequency was significantly higher for women receiving therapies that include trastuzumab, compared to women receiving the same therapies but without trastuzumab.

While estimates of relative risk are heterogeneous between the studies, the pooled estimates generally correspond well with the prior literature on the topic. Our study is based on a larger evidence base than prior literature, and includes analyses of severities and potential subgroup differences. Previously it has been suggested that cardiac effects, neutropenia and susceptibility to infection, skin & nail effects, possible haematological toxicity, diarrhoea and fatigue were likely side effects of trastuzumab alone. Our findings confirm that the risks of congestive heart failure and other cardiac events, susceptibility to infections (particularly respiratory infections), diarrhoea (and other gastrointestinal effects), skin disorders (mainly rashes), and fatigue appear to be increased by using trastuzumab as a treatment for breast cancer. However, interestingly we do not see a significantly increased likelihood of anaemia (the relative risk

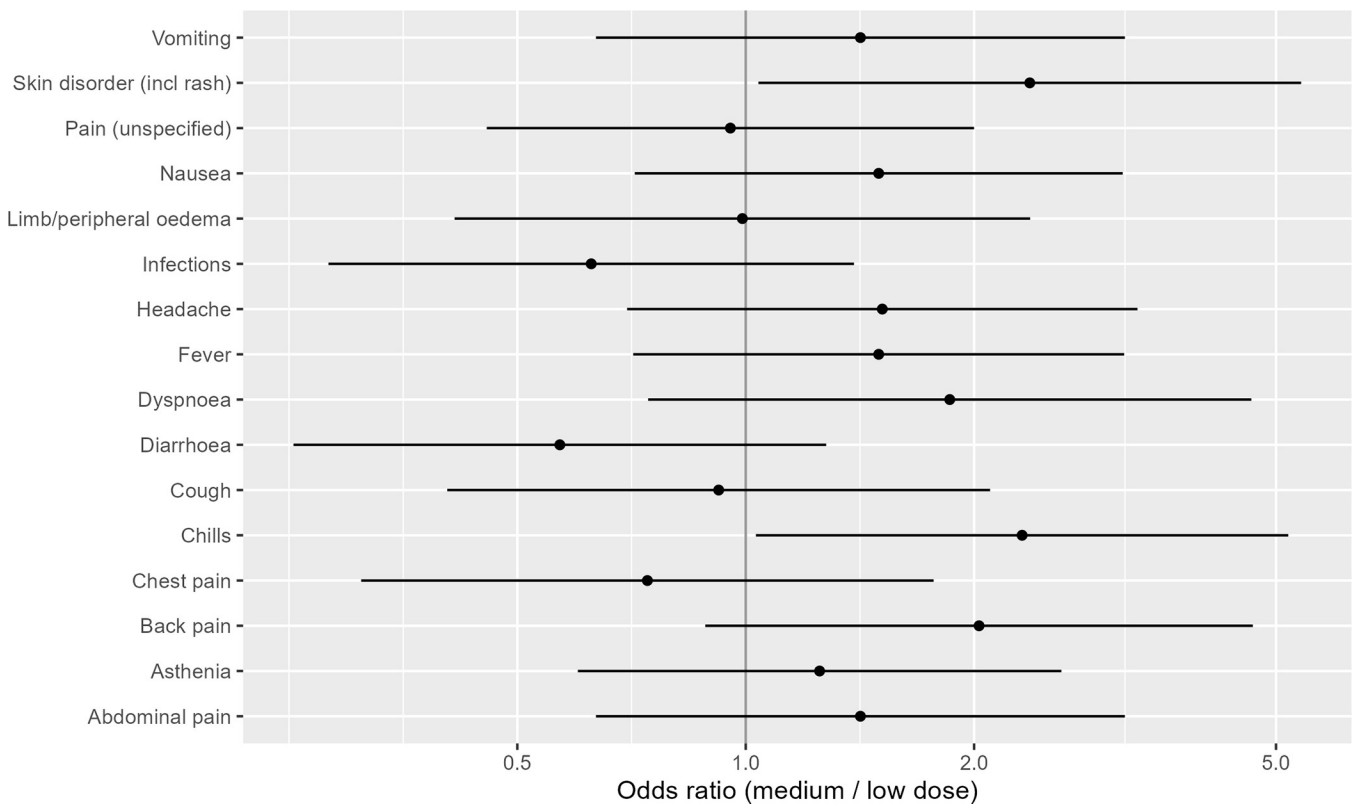

**Fig 5. Odds ratios of adverse events, comparing weekly medium (maintenance dose of 4mg/kg) against low dose (maintenance dose of 2mg/kg) of trastuzumab, from the H0650g trial.**

difference for this was just below our practical significance threshold), lymphopaenia, neutropenia or febrile neutropenia–findings which have had mixed support in previous reviews [2, 7, 10, 12, 58].

In addition to the events previously suspected to be due to trastuzumab alone, we also found a significant increase in the likelihood of back pain and undefined pain, chills, fever, headache, myalgia, asthenia, dyspnoea, dizziness/vertigo, increased lacrimation, nosebleeds, hypertension, and limb/peripheral oedema. Some of these findings are supported by consistent estimates from several studies (see Fig 3A & 3B) and were also listed from clinical trials in FDA paperwork [4]. Some, however, are only reported by a few studies and so are less certain findings, and the majority of the included studies are at risk of bias in some aspect of their collection of adverse event data, especially as all were unblinded (Fig 2). We also illustrated the percentage of the cases in which the side effects were classified as either Serious or Grade 3+ (depending on the classification system used), when such a distinction was made in the data, although this is also open to bias in data recording and reporting.

Investigating whether dose, length of use or concurrent adjuvant therapies affect the side effects of trastuzumab was more difficult. In descriptive analysis, no differential adverse effects of trastuzumab between studies using different trastuzumab doses was apparent, though this comparison was difficult given the extent of heterogeneity in effects and the small number of studies. Only one small trial directly compared two dosing regimes across two arms, without any clear differential adverse effects between different doses. However, strong evidence on the effect of length of trastuzumab use was available from the individual patient-level data from the large HERA trial, since it had arms that were given 1 year and 2 years of treatment, as well

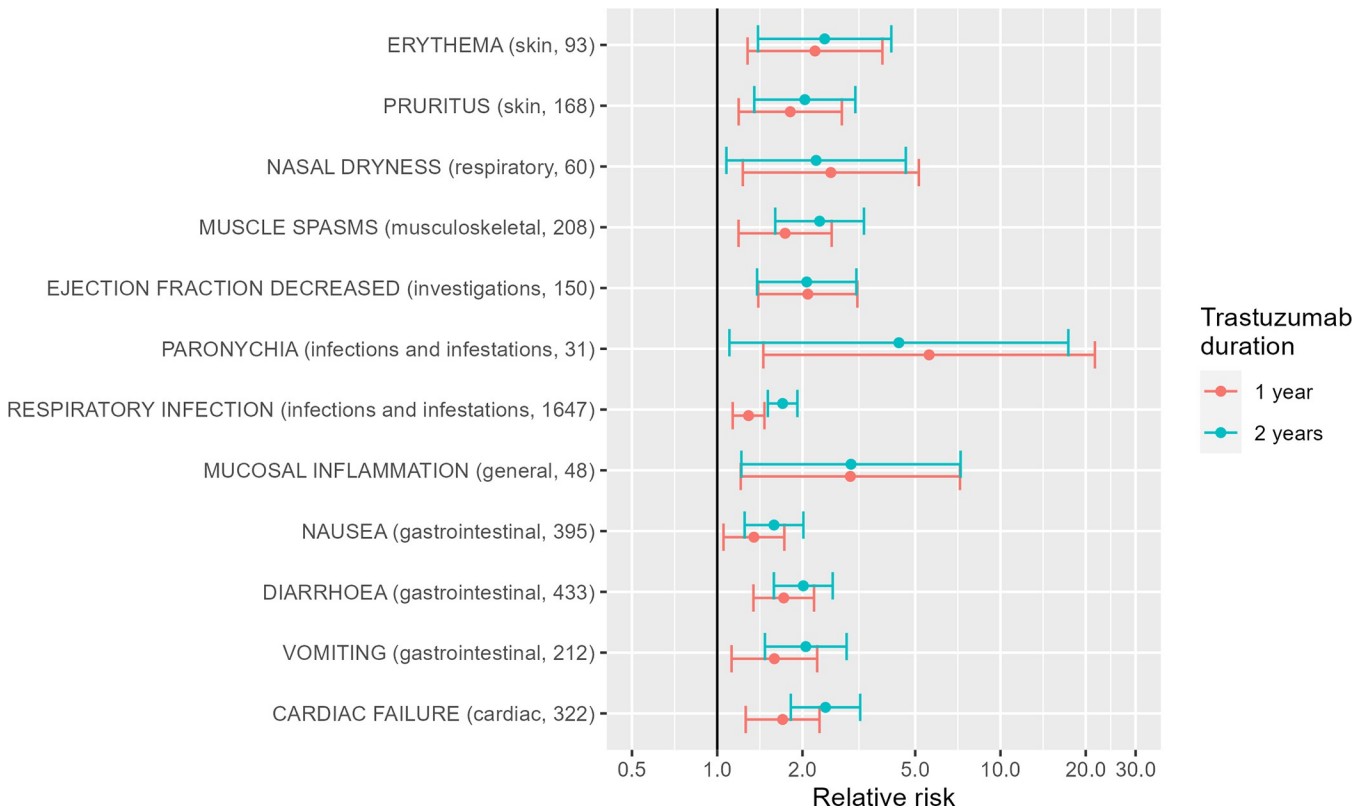

**Fig 6. Adverse effects of trastuzumab, estimated from data from the HERA trial.** Relative risks with 95% confidence intervals, showing only those with statistically significant relative risk > 1.5 or risk difference > 2%. Number of events observed over all three arms shown in brackets.

as a control arm with no trastuzumab. It showed that longer use was associated with more frequent adverse effects of several kinds (see Fig 5), especially respiratory infections, but also gastrointestinal effects, muscle spasms and cardiac failure.

The individual patient-level data is also of use in assessing the potential interactions between trastuzumab and other concurrent drug regimes. However, in the HERA trial for which we had this data, only hormone therapy was given concurrently (chemotherapy courses were mostly completed before trastuzumab therapy). The effect of nausea caused by trastuzumab was higher among women who were ER+/taking hormone therapy. We were therefore unable to add to the existing literature investigating the potential interaction between trastuzumab and chemotherapy relating to cardiac effects, although this has been well studied elsewhere (e.g. [11]). We also did not have the necessary data to investigate whether the adverse effects of trastuzumab are modified by combining trastuzumab with other ERBB2-targetted therapies such as lapatinib or pertuzumab.

To investigate whether the adverse effects of trastuzumab vary by patient characteristics, we analysed the patient-level data from the large HERA trial. This found no interactions, other

**Table 2. Subgroup-specific effect of trastuzumab (taken for 1 or 2 years) on the risk of nausea is shown for each subgroup as an odds ratio, along with the ratio of odds ratios between subgroups with and without the given risk factor.** 291 patients out of the 5,000 in the trial suffered nausea.

| Risk Factor | OR ratio with: without | OR (1 year, without) | OR (2 years, without) | OR (1 year, with) | OR (2 years, with) | P (interaction) |
|---|---|---|---|---|---|---|
| ER+ | 2.01 (1.19–3.4) | 1.18 (0.83, 1.67) | 1.2 (0.85, 1.7) | 1.71 (0.9, 3.26) | 2.41 (1.28, 4.54) | 0.031 |
| Hormone therapy | 1.92 (1.14–3.23) | 1.16 (0.81, 1.66) | 1.21 (0.84, 1.72) | 1.7 (0.89, 3.24) | 2.31 (1.23, 4.35) | 0.049 |

than with ER status (and ER+ women were likely also to be taking hormone therapy, as described above).

This large and systematic study, then, clarifies the suspicions raised by previous reviews–confirming several of the suspected side-effects of trastuzumab alone (cardiac, gastrointestinal, skin rashes, and susceptibility to infections–particularly respiratory as well as fatigue), as well as pinpointing several from the long list of combined effects of trastuzumab and chemotherapy which can now more confidently be assigned as side effects of trastuzumab alone (including pain, asthenia and skin/nail disorders), and give closer estimates as to their likely frequencies.

This study was conducted as a result of requests from patients who told us that they would like more information about the likelihood and likely severity of side effects of therapies when making decisions. Many clinicians and patients currently use the Predict:Breast Cancer website (breast.predict.nhs.uk) to discuss options and potential benefits from adjuvant therapies, because it illustrates quantitative evidence on these benefits. However, patients are not able to be provided with similarly quantified evidence on the potential harms of the therapies to weigh up against these potential benefits as adverse effects have not previously been quantified to the same extent. The numbers derived from this study should provide patients being treated for breast cancer and their healthcare professionals with a resource to source such data on the additional risks of different side effects resulting from adding trastuzumab to their treatment regime. However, there are many limitations to the numbers derived in this study and they must be communicated with adequate uncertainties and combined with clinical experience.

The first major limitation is that although while heterogeneous in many cases, the estimates of relative risk were generally qualitatively consistent. The absolute frequencies of adverse events, however, were very different between different studies, likely as a result of different degrees of reporting or recording of side effects, especially given the open-label nature of the trials (although potentially also from different concurrent treatment regimes being used in different studies). This creates a large amount of statistical uncertainty around the absolute numbers.

However, there are deeper-level uncertainties involved. This study only looked at adverse events recorded during clinical trials, which–as previously noted [56]–vary greatly in their level of recording and reporting of adverse effects, and may not record long-term effects. Here, the longitudinal experience of clinicians who have, over their careers, observed large numbers of patients throughout their treatment, is important to access.

It is hoped that future consistent reporting via registry and data sites such as clinicaltrials.gov will help provide fuller data on adverse effects in the future, allowing more meta-analyses (such as a similar meta-analysis on the effects of trastuzumab when used in gastric cancer patients). We are making all our extracted data freely available in a repository (https://doi.org/10.17605/OSF.IO/ER9FZ) to help facilitate further analyses. It will also be important to combine these clinical-trial-derived data with expert opinions and experience in order to produce a quantitative resource for clinicians and patients which present estimates of adverse effects similar to those currently available which estimate clinical benefits. These sorts of decision support tools can then inform the weighing up of potential risks and benefits in clinical practice to support the choice of the best treatment options for each individual patient. Greater knowledge about what side effects to expect, and what to do when they occur, can also improve management of such symptoms and patient's adherence to a chosen regime [59–61].

## Supporting information

**S1 Checklist. PRISMA 2020 checklist.**
(DOCX)

**S1 File. Characteristics of the trials identified in search.**
(DOCX)

## Acknowledgments

This publication includes research using data from the HERA trial, provided by data contributors Roche, that has been made available through Vivli, Inc. Vivli has not contributed to or approved, and is not in any way responsible for, the contents of this publication.

## Author Contributions

**Conceptualization:** Alexandra L. J. Freeman.

**Data curation:** Christopher Jackson, Leila Finikarides, Alexandra L. J. Freeman.

**Formal analysis:** Christopher Jackson.

**Methodology:** Christopher Jackson, Alexandra L. J. Freeman.

**Visualization:** Christopher Jackson.

**Writing – original draft:** Christopher Jackson, Alexandra L. J. Freeman.

**Writing – review & editing:** Christopher Jackson, Leila Finikarides, Alexandra L. J. Freeman.

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
