## [Decision Letter · Decision Letter 0]

19 Aug 2022

PONE-D-22-21070The adverse effects of trastuzumab-containing regimes as a therapy in breast cancer: a piggy-back systematic review and meta-analysisPLOS ONE

Dear Dr. Freeman,

Thank you for submitting your manuscript to PLOS ONE. After careful consideration, we feel that it has merit but does not fully meet PLOS ONE’s publication criteria as it currently stands. Therefore, we invite you to submit a revised version of the manuscript that addresses the points raised during the review process.

We look forward to receiving your revised manuscript.

Kind regards,

Alessandro Rizzo

Academic Editor

PLOS ONE

Journal Requirements:

Reviewers' comments:

Reviewer's Responses to Questions

**Comments to the Author**

1. Is the manuscript technically sound, and do the data support the conclusions?

Reviewer #1: Partly

2. Has the statistical analysis been performed appropriately and rigorously? 

Reviewer #1: Yes

3. Have the authors made all data underlying the findings in their manuscript fully available?

Reviewer #1: Yes

4. Is the manuscript presented in an intelligible fashion and written in standard English?

Reviewer #1: Yes

5. Review Comments to the Author

Reviewer #1: Breast cancer (BC), the most commonly diagnosed cancer in women worldwide, is a heterogeneous disease, comprising distinct biologic entities with different prognosis and oncogenic drivers. Four primary clinical subtypes of BC: luminal A-like, luminal B-like, human epidermal growth factor receptor 2 (HER2) positive, and triple negative BC (TNBC) are described. Historically, the HER2-positive subgroup had a worse prognosis. The development of agents targeting HER2 has provided significant clinical benefits, changing its natural history.

The study addresses a very important, timely topic in breast cancer management.

Some changes are required before eventual publication:

- the authors conducted a very comprehensive research and should be commended for this. However, the discussion section could be expanded, and the authors should further highlight the limitations of the current paper.

- In addition, a more personal perspective should be included in order to report some comments regarding how this study may impact clinical practice

- The background of medical treatment for HER2 positive disease regardless of tumor site should be further discussed in the introduction section, and some recently published papers added, only for a matter of consistency (PMID: 34534430; PMID: 33916206 )

Major changes needed.

6. PLOS authors have the option to publish the peer review history of their article (what does this mean?). If published, this will include your full peer review and any attached files.

Reviewer #1: No

---

## [Author Response · Author response to Decision Letter 0]

9 Sep 2022

Below we describe our responses to the reviewer’s points:

- the authors conducted a very comprehensive research and should be commended for this. However, the discussion section could be expanded, and the authors should further highlight the limitations of the current paper.

We thank the reviewer for the positive comments. We have now expended the discussion section, firstly to contain more information about the limitations:

“However, there are many limitations to the numbers derived in this study and they must be communicated with adequate uncertainties and combined with clinical experience.

The first major limitation is that although while heterogeneous in many cases, the estimates of relative risk were generally qualitatively consistent. The absolute frequencies of adverse events, however, were very different between different studies, likely as a result of different degrees of reporting or recording of side effects, especially given the open-label nature of the trials (although potentially also from different concurrent treatment regimes being used in different studies). This creates a large amount of statistical uncertainty around the absolute numbers.

However, there are deeper-level uncertainties involved. This study only looked at adverse events recorded during clinical trials, which – as previously noted [56] – vary greatly in their level of recording and reporting of adverse effects, and may not record long-term effects. Here, the longitudinal experience of clinicians who have, over their careers, observed large numbers of patients throughout their treatment, is important to access.”

- In addition, a more personal perspective should be included in order to report some comments regarding how this study may impact clinical practice

We are delighted to have the opportunity to do so. We have added two sections to the discussion:

“This study was conducted as a result of requests from patients who told us that they would like more information about the likelihood and likely severity of side effects of therapies when making decisions. Many clinicians and patients currently use the Predict:Breast Cancer website (breast.predict.nhs.uk) to discuss options and potential benefits from adjuvant therapies, because it illustrates quantitative evidence on these benefits. However, patients are not able to be provided with similarly quantified evidence on the potential harms of the therapies to weigh up against these potential benefits as adverse effects have not previously been quantified to the same extent. The numbers derived from this study should provide patients being treated for breast cancer and their healthcare professionals with a resource to source such data on the additional risks of different side effects resulting from adding trastuzumab to their treatment regime.”

“It is hoped that future consistent reporting via registry and data sites such as clinicaltrials.gov will help provide fuller data on adverse effects in the future, allowing more meta-analyses (such as a similar meta-analysis on the effects of trastuzumab when used in gastric cancer patients). We are making all our extracted data freely available in a repository to help facilitate further analyses. It will also be important to combine these clinical-trial-derived data with expert opinions and experience in order to produce a quantitative resource for clinicians and patients which present estimates of adverse effects similar to those currently available which estimate clinical benefits. These sorts of decision support tools can then inform the weighing up of potential risks and benefits in clinical practice to support the choice of the best treatment options for each individual patient. Greater knowledge about what side effects to expect, and what to do when they occur, can also improve management of such symptoms and patient’s adherence to a chosen regime [59–61].”

- The background of medical treatment for HER2 positive disease regardless of tumor site should be further discussed in the introduction section, and some recently published papers added, only for a matter of consistency (PMID: 34534430; PMID: 33916206 )

We thank the reviewer for this suggestion and have included four new reference to the introduction, giving readers easy access to a review of trastuzumab’s mechanism of action as well as references to the cancers it is currently licensed for use in as well as some of the new research the reviewer refers to in other tumour types:

“work has been ongoing on understanding its mechanism of action[3]. In addition, trastuzumab (and subsequent biosimilar drugs) have been licenced for use not just for patients with ERRB2 positive breast cancer, but also gastric cancer [4]; and its derivatives show promise in other cancers such as ERRB2 positive non-small-cell lung cancer [5] and are being tested in others [6].”

---

## [Editor Report · Decision Letter 1]

14 Sep 2022

The adverse effects of trastuzumab-containing regimes as a therapy in breast cancer: a piggy-back systematic review and meta-analysis

PONE-D-22-21070R1

Dear Dr. Freeman,

We’re pleased to inform you that your manuscript has been judged scientifically suitable for publication and will be formally accepted for publication once it meets all outstanding technical requirements.

Kind regards,

Alessandro Rizzo

Academic Editor

PLOS ONE

---

## [Editor Report · Acceptance letter]

22 Nov 2022

PONE-D-22-21070R1 

The adverse effects of trastuzumab-containing regimes as a therapy in breast cancer: a piggy-back systematic review and meta-analysis 

Dear Dr. Freeman:

I'm pleased to inform you that your manuscript has been deemed suitable for publication in PLOS ONE. Congratulations! Your manuscript is now with our production department. 

Kind regards, 

on behalf of

Dr. Alessandro Rizzo 

Academic Editor

PLOS ONE